# Effects of older age on contraction-induced intramyocellular acidosis and inorganic phosphate accumulation *in vivo*: A systematic review and meta-analysis

**Luke R. Arieta[1], Zoe H. Smith[1], Amanda E. Paluch[1,2], Jane A. Kent[1]***

**1** Department of Kinesiology, University of Massachusetts, Amherst, MA, United States of America,
**2** Institute for Applied Life Sciences, University of Massachusetts, Amherst, MA, United States of America

* jkent@umass.edu

**Editor:** Daniel Boullosa, Universidad de León Facultad de la Ciencias de la Actividad Física y el Deporte: Universidad de Leon Facultad de la Ciencias de la Actividad Fisica y el Deporte, SPAIN

## Abstract

Although it is clear that the bioenergetic basis of skeletal muscle fatigue (transient decrease in peak torque or power in response to contraction) involves intramyocellular acidosis (decreased pH) and accumulation of inorganic phosphate (Pi) in response to the increased energy demand of contractions, the effects of old age on the build-up of these metabolites has not been evaluated systematically. The purpose of this study was to compare pH and [Pi] in young (18–45 yr) and older (55+ yr) human skeletal muscle *in vivo* at the end of standardized contraction protocols. Full study details were prospectively registered on PROSPERO (CRD42022348972). PubMed, Web of Science, and SPORTDiscus databases were systematically searched and returned 12 articles that fit the inclusion criteria for the meta-analysis. Participant characteristics, contraction mode (isometric, dynamic), and final pH and [Pi] were extracted. A random-effects model was used to calculate the mean difference (MD) and 95% confidence interval (CI) for pH and [Pi] across age groups. A subgroup analysis for contraction mode was also performed. Young muscle acidified more than older muscle (MD = -0.12 pH; 95%CI = -0.18,-0.06; $p$<0.01). There was no overall difference by age in final [Pi] (MD = 2.14 mM; 95%CI = -0.29,4.57; $p$ = 0.08), although sensitivity analysis revealed that removing one study resulted in greater [Pi] in young than older muscle (MD = 3.24 mM; 95%CI = 1.72,4.76; $p$<0.01). Contraction mode moderated these effects ($p$ = 0.02) such that young muscle acidified (MD = -0.19 pH; 95%CI = -0.27,-0.11; $p$<0.01) and accumulated Pi (MD = 4.69 mM; 95%CI = 2.79,6.59; $p$<0.01) more than older muscle during isometric, but not dynamic, contractions. The smaller energetic perturbation in older muscle indicated by these analyses is consistent with its relatively greater use of oxidative energy production. During dynamic contractions, elimination of this greater reliance on oxidative energy production and consequently lower metabolite accumulations in older muscle may be important for understanding task-specific, age-related differences in fatigue.

**Data Availability Statement:** All relevant data are within the manuscript and its Supporting Information files.

**Funding:** This study was funded by the National Institute on Aging, RO1 AG058607, to J. Kent.

**Competing interests:** The authors have declared that no competing interests exist.

## Introduction

Old age is accompanied by a loss of muscle size and strength and, as a result, a decline in physical function [1, 2]. Skeletal muscle fatigue, defined as the transient decrease in muscle force or power in response to contractile activity, can exacerbate this age-related decline in physical function [3, 4] and potentially limit the ability of older adults to complete daily tasks. A full understanding of the mechanisms responsible for age-related differences in muscle fatigue is yet to be established, but is crucial for mitigating the impact of lower physical function in older adults.

Muscle fatigue develops to different degrees in older and younger muscle depending in part on contraction mode [5–8]. During isometric contractions, older adults develop less muscle fatigue than young adults [8–11]. In contrast, during both isokinetic (i.e., constant velocity) and isotonic (i.e., constant load) dynamic contractions, the lesser fatigue of older muscle is generally abolished. Some studies show no difference in fatigue between younger and older muscle following dynamic contractions [8, 9, 12], while others report greater fatigue in older muscle [8, 13–15]. First quantified by Christie et al. [5] and recently revaluated with newer data [7], the way in which fatigue is measured in response to dynamic contractions, either as muscle torque or power, likely contributes to the equivocal results in the literature to date. It appears that, when muscle fatigue is operationally defined and measured as the decline in power, older muscle fatigues to a greater extent than younger muscle [5, 7], which is particularly evident during high-velocity dynamic contractions [8].

The direct energy for muscular work is provided by ATP, which is replenished in the cytosol by the creatine kinase reaction and glycolysis, collectively termed the non-oxidative energy pathways, as well as by oxidative phosphorylation in the mitochondria, which is often referred to as the oxidative energy pathway. An increase in non-oxidative ATP production is accompanied by intramyocellular acidosis (decline in pH) and the accumulation of inorganic phosphate (Pi) as contractions continue [16]. Elimination of oxidative ATP production with ischemia magnifies this shift in energy production to the non-oxidative pathways [17], with the consequence of greater intramyocellular metabolite accumulation. Somewhat surprisingly, a systematic review and meta-analysis found that, overall, older muscle has a greater capacity to produce ATP *in vivo* via the oxidative pathway compared with younger muscle [18]. This greater mitochondrial oxidative capacity of older muscle could allow a relative sparing of the need for non-oxidative ATP production during standardized contraction protocols, and thus reduce acidosis and the accumulation of Pi, particularly during high-intensity contractions that likely recruit all fiber types [19, 20].

Regardless of age, the preponderance of evidence indicates that intramyocellular acidosis and increased [Pi] are causal mechanisms of muscle fatigue. This concept has been established at the molecular [21–24], cellular [25–27] and *in vivo* [10, 14, 17, 28, 29] levels, and confirmed *in silico* [30]. Thus, the bioenergetic basis of skeletal muscle fatigue, especially in response to high-intensity muscle contractions, has robust support in the literature. Diprotonated inorganic phosphate ($H_2PO_4^-$) becomes the dominant inorganic phosphate species when pH < 6.75, and thus reflects changes in both [Pi] and pH in the cell. As a result, relationships between muscle fatigue and [$H_2PO_4^-$] also have been established [10, 14, 17]. Under various conditions, changes in [ATP], [adenosine diphosphate] (ADP), [glycogen], electrolytes (e.g., sodium and potassium) and reactive oxygen species, as well as neural mechanisms (e.g., motor unit activation) and excitation-contraction coupling, may also contribute to muscle fatigue [16]. With this concept of the bioenergetic basis of muscle fatigue in mind, the existing literature on the effects of old age on muscle fatigue- absent measures of intramyocellular energetics- suggests that older muscle may acidify less and accumulate less Pi during isometric contractions, with these differences eliminated or reversed in response to dynamic

contractions. However, the current body of evidence regarding age-related differences in muscle acidosis and Pi accumulation in response to isometric and dynamic contractions has not been evaluated systematically.

[31]Phosphorus magnetic resonance spectroscopy (MRS) is a non-invasive tool used to measure intramyocellular pH and Pi *in vivo* during muscular work [31]. Investigators have applied this technique to the study of cellular bioenergetics and metabolite accumulation in young and older populations, but the question remains as to whether there is a consistent, age-related difference in muscle acidosis or Pi accumulation in response to muscle contractions. Additionally, it is not known whether contraction mode moderates any potential age-related differences in metabolite accumulation. Therefore, the purpose of this study was to systematically search the literature and conduct a meta-analysis to determine whether there is an age-related difference in pH or [Pi] in response to human skeletal muscle contractions *in vivo*. Based on the separate bodies of literature about muscle fatigue and oxidative capacity in aging, we hypothesized that there would be no difference in pH or accumulation of Pi between young and older muscle overall, but that contraction mode would moderate this relationship such that older muscle would acidify and accumulate Pi to a *lesser* extent in response to isometric contractions but to a *greater* extent in response to dynamic contractions. As such, these bioenergetic results would mirror the literature regarding muscle fatigue in aging.

## Methods

### Search strategy

This systematic review and meta-analysis was completed in adherence to the Preferred Reporting Items for Systematic Reviews and Meta-Analyses (PRISMA) guidelines (S1 Appendix). PubMed, Web of Science, and SPORTDiscus databases were systematically searched on the 29th of July 2022 and updated on the 15th of June 2023 and 19th of January 2024. Search terms addressing the study population (younger vs. older adults), outcome variables (end of contraction protocol pH and [Pi] measured by [31]phosphorus MRS), and exposure (standardized contraction protocols) were used. The search strategy details are available in S2 Appendix. There were no restrictions placed on search results. The reference lists of all studies selected for inclusion were also searched for studies that may have been missed in the database search. The full study details and search strategy were prospectively registered on PROSPERO (CRD42022348972) prior to formal searching and data extraction.

### Study selection

Search results were uploaded to the online systematic review software Rayyan [32]. Titles and abstracts were then screened by two independent reviewers (LRA and ZHS) and disagreements were resolved by a third (JAK). Cross-sectional studies and baseline data from intervention studies were considered for inclusion. Inclusion criteria were: 1) a direct comparison of young (18–45 yr) and older (55+ yr) human subjects, 2) text available in English, 3) use of a standardized muscle contraction protocol in both age groups (e.g., relative to maximal strength), 4) intramyocellular pH (or the concentration of hydrogen ion, [H$^+$]) measured at the end of the contraction protocol by [31]phosphorus MRS, 5) group mean and measure of variance reported or obtainable upon request, and 6) original research findings. The cutoff for older adults of 55+ years was chosen to allow for inclusion of early studies that often considered older adults to be 55+ years. Studies were included if the mean age of the participants in each group met this criterion.

Exclusion criteria included: 1) contraction protocols successfully designed to limit acidosis (e.g., short, "oxidative capacity" protocols; no studies were eliminated based purely on the

length of the protocol), which worked in tandem with the second exclusion criterion, 2) studies in which mean intramyocellular pH of at least one group was not $\leq 6.90$, which has been established as an appropriate cutoff for muscle acidosis [33], and is consistent with the recommendations in the recent consensus paper for using $^{31}$phosphorus MRS in skeletal muscle [31]. These exclusion criteria were selected to eliminate studies in which the metabolic demands placed on the muscle were not sufficient to elicit a potent energetic response and, in turn, acidosis. Including studies that did not observe muscle pH $\leq 6.90$ could artificially mask age-related differences due to an insufficient perturbation from resting conditions. Studies included for full-text review were then moved to Covidence (Covidence systematic review software, Veritas Health Innovation, Melbourne, Australia), and the full-text review was completed by the same two reviewers with disagreements resolved by the same third reviewer.

## Data extraction

Data extraction was completed independently in Covidence by two reviewers (LRA and ZHS). A custom-written data extraction template was used in the Extraction 2 tool. Extracted data included: sample size, sex distribution of study groups, contraction protocol details (i.e., muscle group, isometric vs. dynamic, intensity, duty cycle, and duration), and the means and measures of variance for age, and pH and [Pi] during the final portion of the contraction protocol. As an exploratory analysis, the means and measures of variance for [$H_2PO_4^-$] were also extracted or calculated from individual pH and [Pi] data [34], when available. Measures of variance were converted to standard deviation, as needed [35]. The data extracted by each reviewer were compared, and disagreements were resolved by agreement from both reviewers. Requests for missing data were sent to corresponding authors and one study was excluded because missing data could not be obtained. In the case of two studies for which data were only available graphically, WebPlotDigitizer (Version 4.6, Pacifica, California) was used to extract numerical data for Pi [36] or pH and Pi [10].

In some cases, multiple contraction protocols from the same group of participants were included as these unique effects were considered informative to the overall results. In studies for which a protocol was repeated with conditions intentionally altering normal bioenergetic responses to muscle contractions (e.g., ischemia or nutritional supplementation), the "non-control" condition was excluded. In the case of multiple studies based on the same group of participants with the same contraction protocol, only the first study published was selected for inclusion in the meta-analysis.

## Quality assessment

A modified version of the Newcastle-Ottawa Quality Assessment Scale [37] adapted for cross-sectional studies was used to assess the quality of the included studies (S3 Appendix) [7]. This scale evaluates the quality of studies based on a total of nine stars across three sections: selection (four stars), comparability (two stars), and outcome (three stars). Two reviewers (LRA, ZHS) rated the quality of each included study, independently. Disagreements were discussed between the two reviewers and resolved by agreement from both reviewers. A risk of bias analysis [38] was completed that rated each study in each section (i.e., selection, comparability, and outcome) as "good", "fair", or "poor" (S1 Table).

## Statistics

Meta-analytic statistics were completed in RStudio (version 2023.09.1+494) using the meta package (version 6.2–1 [39]). A random-effects model was used to calculate mean differences (MD) and 95% confidence intervals (CI) for pH and [Pi] between young and older groups. A

negative value represented lower pH or Pi in younger muscle and a positive value represented lower pH or Pi in older muscle. The influence of contraction mode (isometric or dynamic) on final pH and [Pi] was evaluated with subgroup analyses. A test for subgroup differences was then used to determine whether contraction mode was a significant moderator of this effect. Probability values of <0.05 were considered statistically significant. Group mean ± SD for the young and older groups for pH and [Pi] were also calculated using the equation in Table 6.5.a of the Cochrane Handbook for Systematic Reviews of Interventions [35]. These means ± SD were calculated to provide an estimate of the overall metabolic perturbation in the young and older groups and were not used in any formal statistical analyses; therefore, the difference between these calculated means are not expected to equal the MD from the formal meta-analysis. The same statistical approach was used for the exploratory analysis of [$H_2PO_4^-$].

Between-studies heterogeneity was evaluated with the $I^2$ statistic and interpreted as low (<25%), moderate (25–75%), or high (>75%) heterogeneity [40]. Funnel plots with mean differences on the x-axes and standard errors on the y-axes were created to assess publication bias, along with the Egger's test [41]. Gaps or asymmetries in a funnel plot and a $p$ value <0.05 from the Egger's test suggest potential publication bias. Study quality for each article included was scored as the number of stars from the modified Newcastle-Ottawa Quality Assessment Scale. The total number of stars from each individual study along with the average number and range of stars for all studies was calculated.

## Results

### Database searches and screening

The searches returned 3,128 articles. After removing duplicates, 2,921 articles were screened. Title and abstract screening eliminated 2,798 of these papers, leaving 123 papers for full-text review. Of these, 111 papers were excluded for the following reasons: no direct comparison of young and older muscle (k = 68), short contraction protocols designed to limit acidosis, or at least one group did not have a pH ≤ 6.90 (k = 19), contraction protocol not standardized (k = 15), redundant dataset or study group (k = 3), no contraction protocol completed (k = 2), non-English text (k = 1), not original research (k = 1), not peer reviewed (k = 1), and pH measured at wrong timepoint (e.g., contractions completed outside of magnet) (k = 1). This left 12 studies from the database searches to include in the meta-analysis.

Three of the included studies contained multiple effects. Kent-Braun et al. [10] reported values from men and women separately and both are included as separate effects. Tevald et al. [42] and Fitzgerald et al. [43] included two unique contraction protocols, and both are included as separate effects in the meta-analysis. Additionally, data from Layec et al. [44] and Layec et al. [45] are from a similar cohort, but the contractions protocols are unique so both studies are included as separate effects. Thus, a total of 15 effects were included in the meta-analysis.

The reference lists of the 12 studies included 366 additional unique studies not captured in the initial or follow-up searches. Of these studies, 10 full texts were reviewed and zero were included in the meta-analysis. Complete details of the search results are presented in the PRISMA flow diagram (Fig 1). Table 1 shows the characteristics of the studies included in this analysis. One study [36] reported [Pi] in mmol·kg wet weight,[-1] which was converted to mM [46] and included.

### Overall pH and [Pi]

The mean ± SD pH and [Pi] data at the end of each study's contraction protocol are given in Figs 2 and 3, respectively. At the end of the contraction protocols, pH was lower in younger

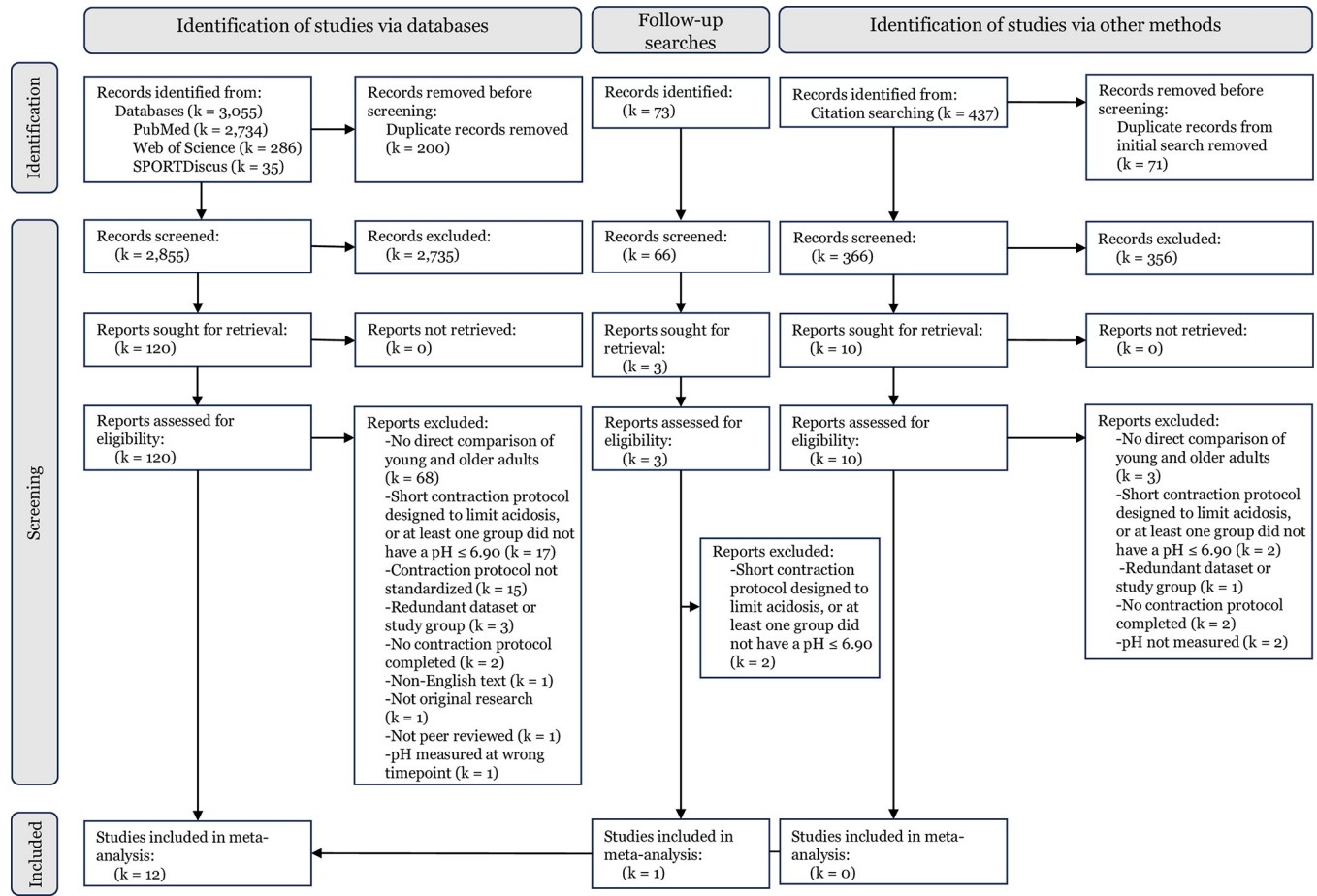

**Fig 1. Preferred Reporting Items for Systematic Reviews and Meta-Analyses (PRISMA) flow diagram.** The process for identification and inclusion of appropriate studies is illustrated.

(mean ± SD, 6.74 ± 0.19 pH) than older (6.86 ± 0.19 pH) muscle (Fig 2). End-exercise intra-myocellular [Pi] tended to be greater in younger (26.20 ± 11.19 mM) than older (24.19 ± 10.56 mM; Fig 3); a sensitivity analysis using the leave-one-out method revealed that when Sundberg et al. [14] was removed from the overall analysis, younger (26.40 ± 11.42 mM) muscle had greater intramyocellular [Pi] compared with older (23.66 ± 10.66 mM) muscle (MD = 3.24 mM; 95% CI = 1.72, 4.76; $p < 0.01$; $I^2$ = 64%; k = 8, 11 effects). Removing the same study from the overall effect for pH did not change those results (MD = -0.14 pH; 95% CI = -0.19, -0.09; $p < 0.01$; $I^2$ = 76%; k = 11; 14 effects). None of the studies included in the meta-analysis reported age-related differences in pH or Pi at rest. Removing the two studies that reported a mean age for the older group of <65 years did not change the overall results for pH (MD = -0.13 pH; 95% CI = -0.19, -0.07; $p < 0.01$; $I^2$ = 87%; k = 10; 13 effects) or [Pi] (MD = 3.24 mM; 95% CI = 1.71, 4.77; $p < 0.01$; $I^2$ = 68%; k = 7; 10 effects).

In the subset of studies for which [$H_2PO_4^-$] could be determined, there was no difference in this metabolite between young (16.92 ± 7.29 mM) and older (13.93 ± 6.69 mM) muscle at the end of contractions (MD = 2.83 mM; 95% CI = -0.40, 6.06; $p$ = 0.09; $I^2$ = 91%; k = 5; 8 effects; S3 Fig). Similar to the case for [Pi], removing the study by Sundberg et al. [14] resulted in greater [$H_2PO_4^-$] in young (17.42 ± 7.37 mM) compared with older (13.43 ± 6.83 mM) muscle (MD = 4.23 mM; 95% CI = 2.11, 6.35; $p < 0.01$; $I^2$ = 81%; k = 4; 7 effects).

**Table 1. Study characteristics.**

| Reference | Age (years) | | Sample Size | | Muscle Group | Contraction Protocol Details | | | |
| | Young | Older | Young (M/F) | Older (M/F) | | Mode | Intensity | Frequency | Duration |
|---|---|---|---|---|---|---|---|---|---|
| Taylor [47] | 20–29 | 70–83 | 20 (N/A) | 6 (N/A) | PF | Dynamic | 10% lean mass for 5 min incremented by 2% every 1.25 min | 0.5 Hz | ~11–12 min* |
| Chilibeck [48] | 27.5 ± 2.0 | 66.9 ± 3.7 | 6/4 | 2/8 | PF | Dynamic | 80% pH threshold | 0.5 Hz | 5 min |
| Smith [36] | 31 ± 5.2 | 58 ± 4.5 | 4/1 | 3/1 | KE | Dynamic | Maximal sustainable resistance for 2 min | $0.61\overline{66}$ Hz | 2 min |
| Kutsuzawa [49] | 28.1 ± 5.0 | 61.4 ± 4.6 | 8/1 | 7/2 | FF | Dynamic | 7% maximum grip strength of non-dominant arm | $0.\overline{33}$ Hz | 3 min |
| Kent-Braun (M) [10] | 33.5 ± 6.5 | 74.4 ± 5.3 | 10/0 | 11/0 | DF | Isometric | 10% MVIC incrementing every 2 min by 10% MVIC | 4 s on, 6 s off | 16 min |
| Kent-Braun (F) [10] | 32.3 ± 4.8 | 75.0 ± 5.9 | 0/10 | 0/10 | DF | Isometric | 10% MVIC incremented every 2 min by 10% MVIC | 4 s on, 6 s off | 16 min |
| Lanza [50] | 22 ± 2.8 | 75 ± 14.1 | 8/0 | 8/0 | DF | Isometric | MVIC | Sustained | 1 min |
| Lanza [17] | 27 ± 4.5 | 70 ± 4.2 | 10/10 | 10/8 | DF | Isometric | MVIC | 12 s on, 12 s off | 132 s |
| Tevald (f50) [42] | 24.7 ± 3.1 | 71.9 ± 4.8 | 7/0 | 8/0 | DF | Isometric | Stimulated | Frequency required to reach 50% MVIC | 90 s |
| Tevald (25 Hz) [42] | 24.7 ± 3.1 | 71.9 ± 4.8 | 7/0 | 8/0 | DF | Isometic | Stimulated | 25 Hz | 60 s |
| Layec [44] | 22.4 ± 1.6 | 72.2 ± 7.3 | 8/9 | 8/9 | PF | Dynamic | 40% work rate max | 1 Hz | 5 min |
| Layec [45] | 22 ± 1.6 | 74 ± 8.1 | 9/9 | 9/9 | PF | Dynamic | 120% work rate max | 1 Hz | 1 min |
| Sundberg [14] | 22.7 ± 1.2 | 76.4 ± 6.0 | 1/6 | 1/7 | KE | Dynamic | 20% MVIC | 0.5 Hz | 4 min |
| Fitzgerald (IsoT) [43] | 27.5 ± 3.9 | 71.2 ± 5.0 | 6/4 | 5/5 | KE | Dynamic | 20% MVIC | 0.5 Hz | 4 min |
| Fitzgerald (IsoK) [43] | 27.5 ± 3.9 | 71.2 ± 5.0 | 6/4 | 5/5 | KE | Dynamic | $120° \cdot s^{-1}$ | 0.5 Hz | 4 min |

Studies are ordered chronologically. 25 Hz = 25 Hz stimulation protocol; DF = ankle dorsiflexors; F = females; f50 = stimulation frequency at 50% of maximal force protocol; FF = forearm flexors; IsoK = isokinetic protocol; IsoT = isotonic protocol; KE = knee extensors; M = males; MOD = moderately active; MVIC = maximal voluntary isometric contraction; PF = ankle plantar flexors; SED = sedentary. *Duration not standardized but not statistically different between groups. Age is reported as mean ± SD or range.

## Effects of contraction mode

Of the 12 studies included, four used isometric contraction protocols (6 effects) and eight used dynamic protocols (9 effects, Table 1). For isometric contractions, pH was lower (Fig 4) and [Pi] greater (Fig 5) in young (pH = 6.76 ± 0.15 pH; [Pi] = 29.24 ± 8.01 mM) compared with older (pH = 6.94 ± 0.11 pH; [Pi] = 25. 39 ± 7.04 mM) muscle. For dynamic contractions, there was no statistical difference between young (pH = 6.73 ± 0.21 pH; [Pi] = 23.39 ± 12.91 mM) and older (pH = 6.81 ± 0.21 pH; [Pi] = 23.06 ± 13.00 mM) muscle for pH (Fig 4) or [Pi] (Fig 5). The test for subgroup differences between contraction modes was significant for pH and [Pi] ($p = 0.02$, both), indicating that contraction mode is a significant moderator of the effect of age on end-contraction pH and [Pi]. The effect of contraction mode on age-related differences in $[H_2PO_4^-]$ was not analyzed due to the small number of studies with available data.

## Publication bias and quality assessment

There was no evidence of publication bias by visual inspection of symmetry in the pH and [Pi] funnel plots (S1 and S2 Figs). Likewise, the Egger's tests for pH ($p = 0.446$) and Pi ($p = 0.122$) were not significant. The quality assessment score for the included studies averaged 5.3 ± 1.5

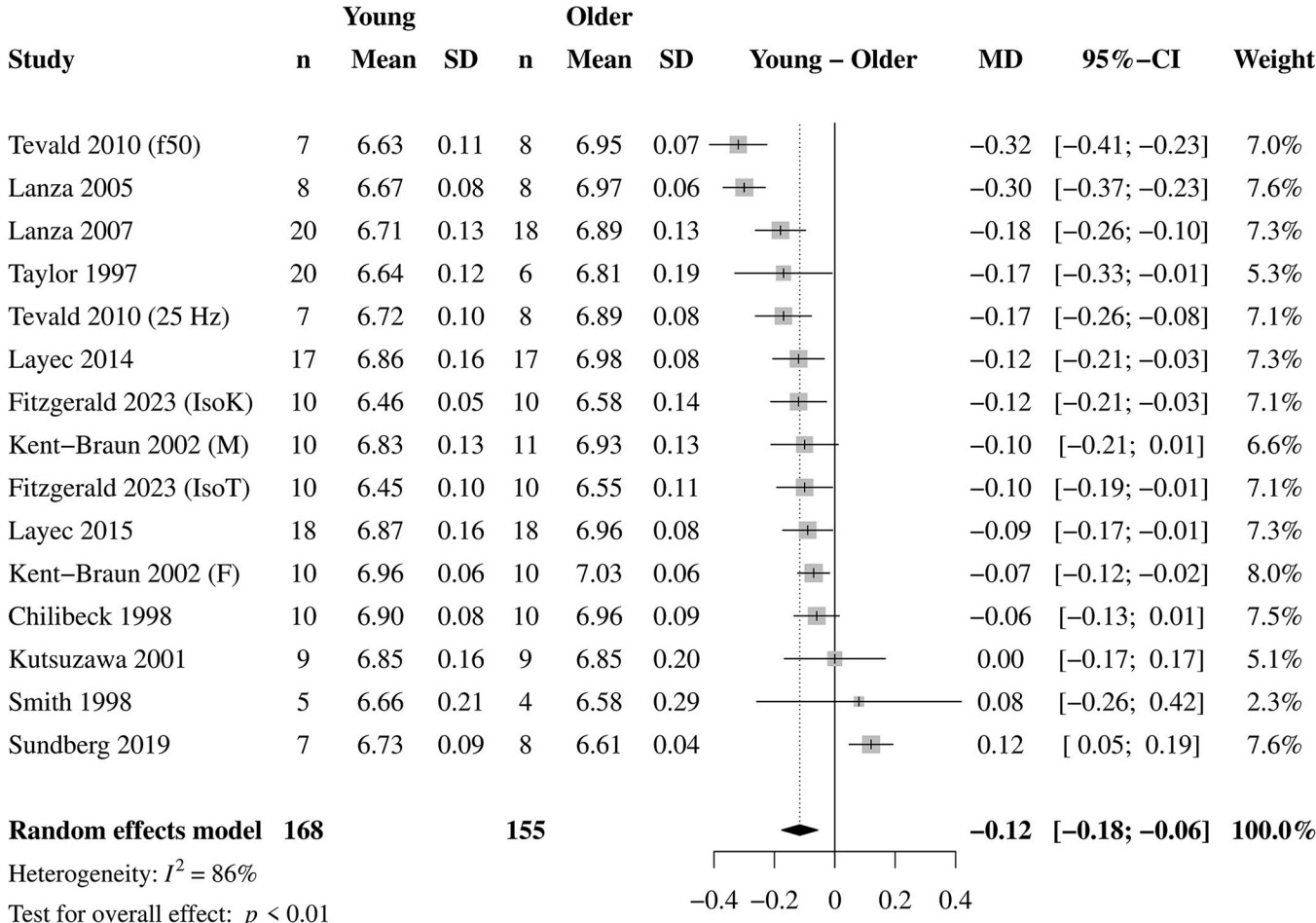

| Study | n | Young Mean | SD | n | Older Mean | SD | Young – Older | MD | 95%–CI | Weight |
|---|---|---|---|---|---|---|---|---|---|---|
| Tevald 2010 (f50) | 7 | 6.63 | 0.11 | 8 | 6.95 | 0.07 | | −0.32 | [−0.41; −0.23] | 7.0% |
| Lanza 2005 | 8 | 6.67 | 0.08 | 8 | 6.97 | 0.06 | | −0.30 | [−0.37; −0.23] | 7.6% |
| Lanza 2007 | 20 | 6.71 | 0.13 | 18 | 6.89 | 0.13 | | −0.18 | [−0.26; −0.10] | 7.3% |
| Taylor 1997 | 20 | 6.64 | 0.12 | 6 | 6.81 | 0.19 | | −0.17 | [−0.33; −0.01] | 5.3% |
| Tevald 2010 (25 Hz) | 7 | 6.72 | 0.10 | 8 | 6.89 | 0.08 | | −0.17 | [−0.26; −0.08] | 7.1% |
| Layec 2014 | 17 | 6.86 | 0.16 | 17 | 6.98 | 0.08 | | −0.12 | [−0.21; −0.03] | 7.3% |
| Fitzgerald 2023 (IsoK) | 10 | 6.46 | 0.05 | 10 | 6.58 | 0.14 | | −0.12 | [−0.21; −0.03] | 7.1% |
| Kent−Braun 2002 (M) | 10 | 6.83 | 0.13 | 11 | 6.93 | 0.13 | | −0.10 | [−0.21; 0.01] | 6.6% |
| Fitzgerald 2023 (IsoT) | 10 | 6.45 | 0.10 | 10 | 6.55 | 0.11 | | −0.10 | [−0.19; −0.01] | 7.1% |
| Layec 2015 | 18 | 6.87 | 0.16 | 18 | 6.96 | 0.08 | | −0.09 | [−0.17; −0.01] | 7.3% |
| Kent−Braun 2002 (F) | 10 | 6.96 | 0.06 | 10 | 7.03 | 0.06 | | −0.07 | [−0.12; −0.02] | 8.0% |
| Chilibeck 1998 | 10 | 6.90 | 0.08 | 10 | 6.96 | 0.09 | | −0.06 | [−0.13; 0.01] | 7.5% |
| Kutsuzawa 2001 | 9 | 6.85 | 0.16 | 9 | 6.85 | 0.20 | | 0.00 | [−0.17; 0.17] | 5.1% |
| Smith 1998 | 5 | 6.66 | 0.21 | 4 | 6.58 | 0.29 | | 0.08 | [−0.26; 0.42] | 2.3% |
| Sundberg 2019 | 7 | 6.73 | 0.09 | 8 | 6.61 | 0.04 | | 0.12 | [ 0.05; 0.19] | 7.6% |
| **Random effects model** | **168** | | | **155** | | | | **−0.12** | **[−0.18; −0.06]** | **100.0%** |

Heterogeneity: $I^2 = 86\%$

Test for overall effect: $p < 0.01$

$-0.4 \quad -0.2 \quad 0 \quad 0.2 \quad 0.4$

**Fig 2. Overall effects for pH.** Forest plot for the overall effect of age on intramyocellular pH in response to standardized muscle contractions (k = 12, 15 effects). A negative value represents more acidosis in young compared with older muscle. Overall, pH was 0.12 units lower in young compared with older muscle at the end of the contraction protocols.

(mean ± SD) and ranged from 2 to 7. For the risk of bias analysis of studies, 58% were rated as poor and 42% were rated as fair in the Selection section; 33% were rated as poor, 25% were rated as fair, and 42% were rated as good in the Comparability section; and 100% were rated as good in the Outcome section. Ratings for individual studies are shown in S1 Table.

## Discussion

The purpose of this study was to systematically search the literature and analyze the available data to determine whether there are consistent, age-related differences in intramyocellular pH and [Pi] *in vivo* in response to human skeletal muscle contractions. Our results indicate that, overall, the muscle of young adults develops more acidosis compared with older muscle in response to standardized muscular activity. The results for [Pi] were more nuanced; there was no significant difference in [Pi] across age groups unless the outlier result [14] was removed, which then indicated significantly greater [Pi] in young compared with older muscle. Intra-myocellular pH was lower and [Pi] greater in young muscle compared with old in response to isometric contractions, but these age-related differences in pH and [Pi] were abolished in response to dynamic contractions. The result that the bioenergetic response to muscular

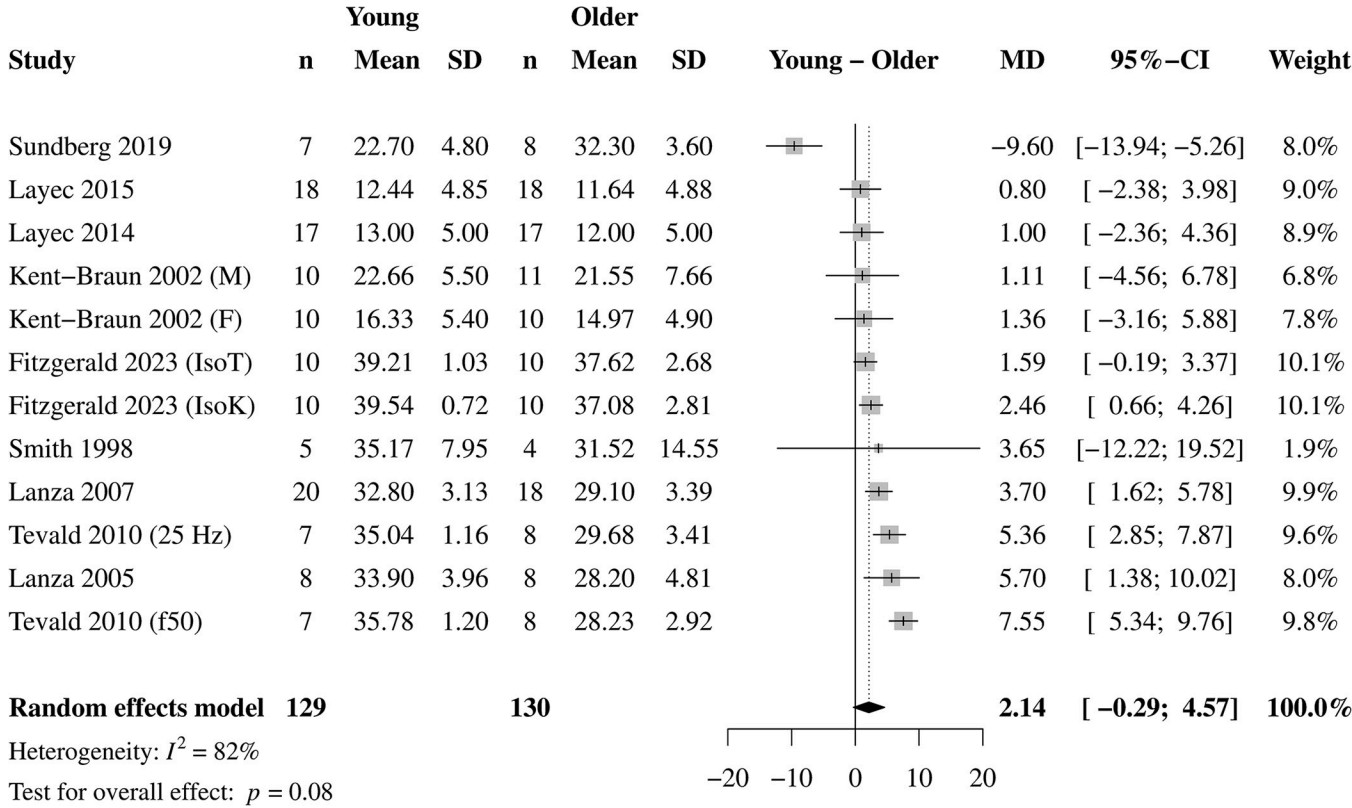

**Fig 3. Overall effects for inorganic phosphate.** Forest plot for the overall effect of age on intramyocellular inorganic phosphate (Pi) in response to standardized muscle contractions (k = 9, 12 effects). A positive value represents greater [Pi] in young compared with older muscle. Overall, there was no difference in end-exercise [Pi] between young and older muscle. However, a sensitivity analysis using the leave-one-out method revealed that when Sundberg et al. [14] was removed, younger muscle had greater intramyocellular [Pi] compared with older muscle (MD = 3.24 mM; 95% CI = 1.72, 4.76; $p < 0.01$; $I^2$ = 64%; k = 8, 11 effects).

contractions is moderated by contraction mode is consistent with the enhanced fatigue resistance reported in older muscle during isometric contractions vs. the equivocal results regarding fatigue in studies that used dynamic contractions. To our knowledge, this is the first systematic evaluation of age-related bioenergetic responses to standardized muscular contractions. These results aid in the interpretation of age-related differences in muscle energetics and fatigue, and are consistent with the role of the metabolic by-products of non-oxidative ATP production in the development of fatigue.

## Age-related differences in pH and [Pi]

Overall, we found that young muscle acidifies 0.12 pH units more than older muscle in response to standardized contraction protocols, with no significant difference for [Pi] (Figs 2 and 3). Additionally, results from the exploratory analysis in a small set of studies indicated no significant age-related difference in [$H_2PO_4^-$] in response to standardized contractions. Notably, removing the study by Sundberg et al. [14] resulted in significant overall effects for [Pi] (and [$H_2PO_4^-$]), such that [Pi] was 3.24 mM and [$H_2PO_4^-$] was 4.23 mM greater in young than older muscle in response to a standardized contraction protocol. The impact of a single study on the results for [Pi] and [$H_2PO_4^-$] is likely due to the significant negative effects (MD [Pi] = -9.60, Fig 3, and MD [$H_2PO_4^-$] = -6.90, S3 Fig) of this study, which is the only study to date to show greater [Pi] and [$H_2PO_4^-$] in older compared with young muscle. Likewise, this was the

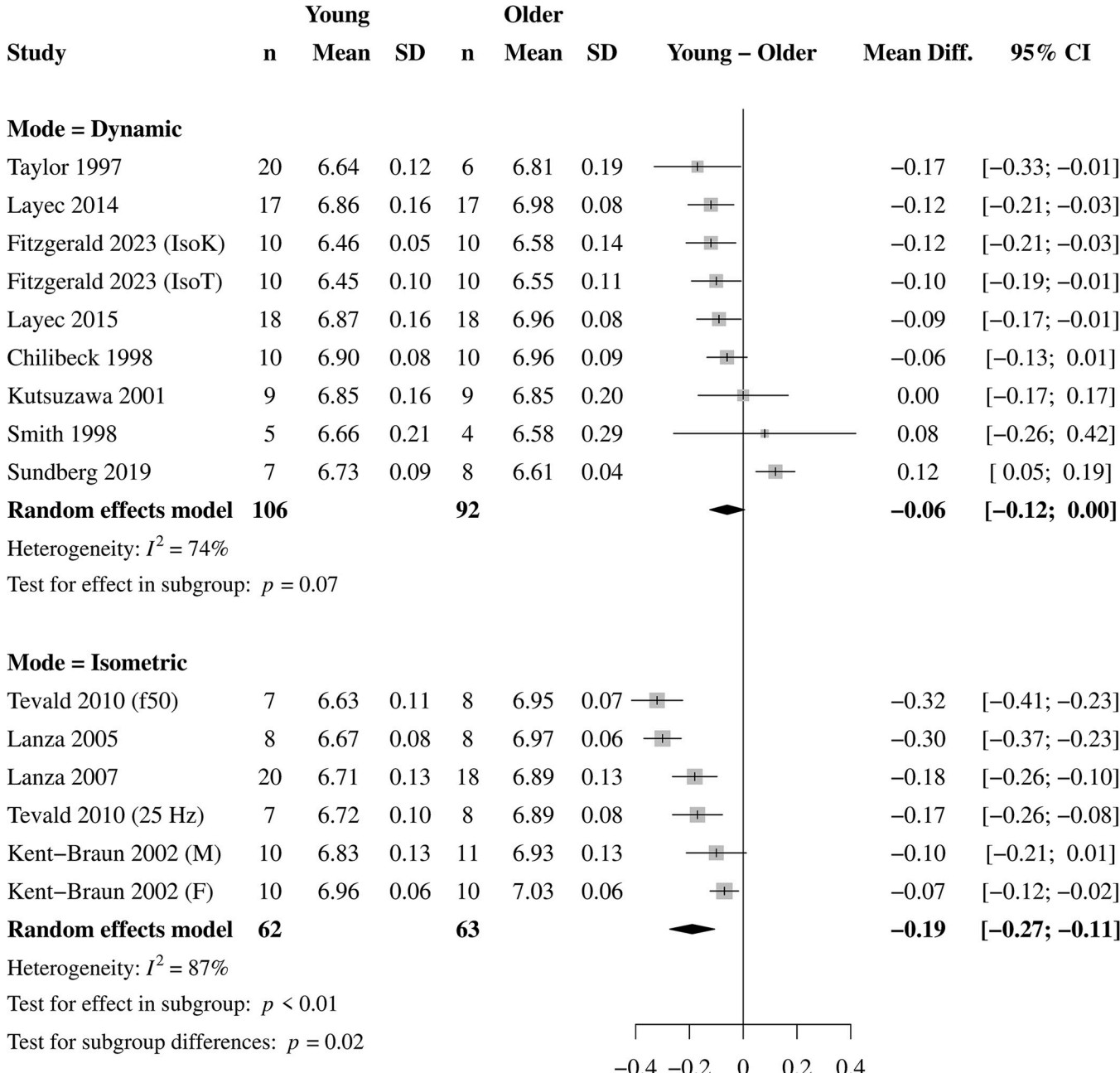

**Fig 4. Effect of contraction mode on pH.** Forest plot for the subgroup analysis on contraction mode on the age-related difference in intramyocellular pH in response to standardized muscle contractions. A negative value represents more acidosis in young muscle. There was no age-related difference in pH in response to dynamic contractions, although this neared statistical significance. In contrast, pH was 0.19 units lower in young compared with older muscle in response to isometric contractions. The test for subgroup differences was significant ($p = 0.02$), indicating that contraction mode moderates the age-related difference in contraction-induced acidosis.

only study to report significantly more acidosis (MD = 0.12, Fig 2) in older compared with young muscle. It is possible that the advanced age (mean 76 yr) of the older group, use of the mixed fiber-type vastus lateralis muscle, the isotonic contraction protocol at 20% of maximal voluntary isometric contraction torque, or a combination of these factors contributed to their unique results.

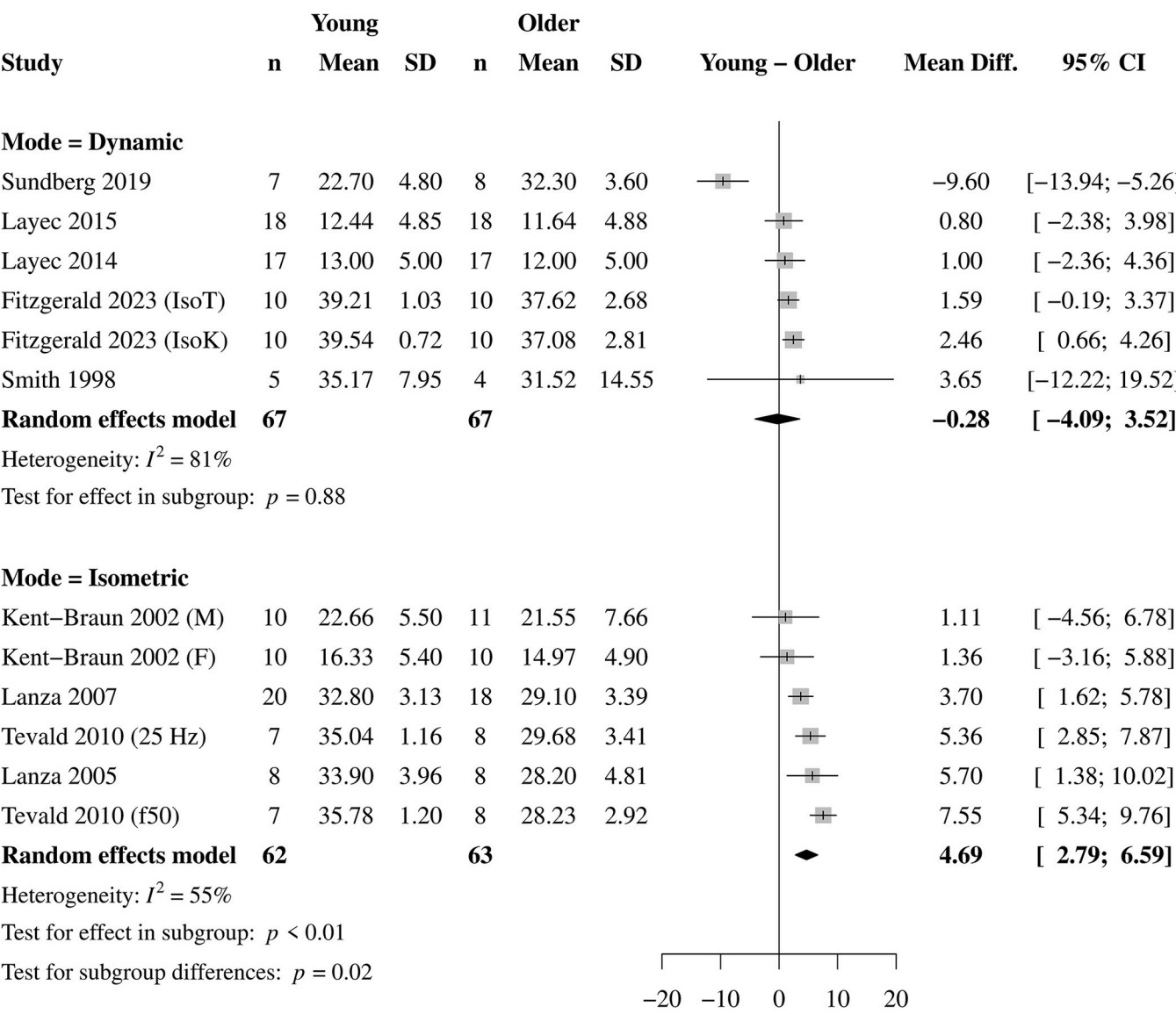

**Fig 5. Effect of contraction mode on inorganic phosphate.** Forest plot for the subgroup analysis on contraction mode on the age-related difference in intramyocellular inorganic phosphate (Pi) in response to standardized muscle contractions. A positive value represents a greater [Pi] in young compared with older muscle. There was no age-related difference in [Pi] in response to dynamic contractions, but [Pi] was 4.69 mM greater in young compared with older muscle in response to isometric contractions. The test for subgroup differences was significant ($p = 0.02$), indicating that contraction mode moderates the age-related difference in contraction-induced accumulation of Pi.

However, a recent study of ~71-year olds by Fitzgerald et al. [43] used the same contraction protocol, which resulted in greater acidosis in young muscle (MD = -0.10) and no difference in [Pi] compared with older muscle. A key difference between these studies was the position of the participants, prone vs. supine for Sundberg et al. [14] and Fitzgerald et al. [43], respectively, which might explain the different results. It is also possible that the prone position compressed feed vessels to the anterior thigh muscles more in the older group, resulting in insufficient blood flow and oxygen delivery. Inadequate oxygen supply would shift ATP production to non-oxidative pathways resulting in lower pH and greater [Pi], similar to the ischemic condition induced by cuff occlusion in the study by Lanza et al. [17]. More studies are needed to

resolve these discrepancies in muscle fatigue and energetic responses, particularly during energetically-costly dynamic contractions in various muscle groups.

The results from this systematic review and meta-analysis suggest that the greater pH and generally lower [Pi] in older muscle in response to muscular contractions likely reflects a greater use of oxidative energy production in older muscle. Greater oxidative production of ATP could enhance fatigue resistance in aging by minimizing the accumulation of these fatigue-inducing metabolites. This result is in agreement with the overall greater oxidative capacity in older muscle [18] and is consistent with the lack of a *de facto* decrease in mitochondrial energy production as a consequence of older age [51].

Additionally, older muscle may activate glycolysis to a lesser extent than younger muscle. Glycolysis is controlled by a dual-control mechanism where both feed-forward (e.g., calcium ions) and feedback (e.g., Pi, adenosine diphosphate, and adenosine monophosphate) mechanisms regulate flux through the pathway [52, 53]. Older muscle may have lower maximal motor unit discharge rates [54, 55], which would reduce the amount of calcium released from the sarcoplasmic reticulum during contractions [56]. Although speculative, lower cytosolic [calcium] in older compared with younger muscle would decrease glycogen breakdown and glycolytic flux, potentially resulting in a blunted activation of the glycolytic pathway in aging and subsequent lesser acidosis. Changes in lactate dehydrogenase activity [57] have been reported in older skeletal muscle, which could influence the fate of glucose and thus acidosis in working muscle. Finally, selective atrophy of fast type II muscle fibers in older muscle [27, 58, 59] may shift energy production to oxidative pathways, sparing the accumulation of fatiguing metabolites.

The relatively high ($I^2$ = 82–86%, all studies) heterogeneity in the overall effects reported here is likely due to the inclusion of studies that involved various contraction protocols, muscle groups, and participant health and physical activity status. The influence of contraction mode (dynamic vs. isometric) is discussed below. Other aspects of the contraction protocol (e.g., intensity, frequency, duration of the contractions) could likewise influence motor unit recruitment, as lower-intensity contractions do not fully recruit all high-threshold motor units that generally innervate type II muscle fibers. A difference in fiber type recruitment could ultimately lead to different energetic pathway use [19, 20, 60] and therefore different levels of fatiguing metabolites. While these issues likely contributed to the high heterogeneity between studies, within each study young and older groups completed the same protocol, making evaluating age-related differences possible. Finally, the different functional, morphological, contractile and energetic characteristics of the various studies used in this analysis certainly contributes to the observed heterogeneity. Additional work is needed to determine the conditions and mechanisms for the relatively greater reliance on oxidative energy production observed in healthy older skeletal muscle.

## Influence of contraction mode

Contraction mode influenced the age-related difference in metabolite accumulation, such that pH was 0.19 greater and [Pi] was 4.69 mM lower in older compared with young muscle in response to isometric contractions, which confirmed our hypothesis. Notably, this age-related difference was eliminated in response to dynamic contractions. Indeed, contraction mode was a significant moderator of the age-related difference in acidosis and [Pi] in response to muscular contractions. It may be that older muscle experiences an energetic "advantage" of a greater reliance on oxidative energy production, resulting in less accumulation of fatiguing metabolites, in response to isometric contractions that is eliminated in response to dynamic contractions. These results are consistent with the roles of pH and Pi in the fatigue process in working

muscle [10, 14, 17], and appear to explain the effects of contraction mode on fatigue in aging muscle.

**Isometric contractions.** In response to isometric contractions, it has long been established that older muscle has greater fatigue resistance than young [8–11], which is in accordance with the greater pH and lower [Pi] observed here in older muscle. A computational model of neuromuscular fatigue confirmed that a lower reliance on glycolytic energy production (i.e., greater pH) is the single greatest factor explaining the fatigue resistance of older muscle during isometric contractions [30]. While there are only four studies (6 effects) to date evaluating age-related differences in the accumulation of these metabolites following standardized isometric contractions, together with the large effect sizes for greater fatigue in young adults in response to isometric contractions from previous meta-analyses [5, 7], there is a consistency across the fatigue and energetic literature suggesting that older muscle experiences a smaller metabolic perturbation than young that may accompany their lower fatigue in response to isometric contractions. The moderate-to-high heterogeneity (55–87%) in the analysis of the isometric contraction protocols is likely due to the use of different contraction intensity, frequency, and durations across the isometric protocols (Table 1), which would yield a variety of energetic responses.

There are several potential mechanisms for the lower accumulation of fatiguing metabolites in response to isometric contractions in older muscle. For example, the older neuromuscular system may undergo selective atrophy of type II muscle fibers [27, 58, 59], which generally rely on non-oxidative energy production to a greater extent than oxidative type I fibers [60], in part due to differences in lactate dehydrogenase isoform, mitochondrial content, and capillarization. However, based on the relative volume of type I and II fibers in young and older muscle, in combination with fiber-type specific phosphofructokinase activities, Lanza et al. [17] concluded that an age-related fiber-type shift could account for only an ~11% decline in phosphofructokinase activity (a proxy for glycolytic flux) and thus could not fully explain the lower use of non-oxidative energy production in older compared with younger muscle observed in that study.

**Dynamic contractions.** Our hypothesis that older muscle would acidify more and accumulate more Pi compared with young muscle in response to dynamic contractions was not supported. However, we did find that the lesser acidosis and [Pi] in older compared with young muscle in response to isometric contractions was eliminated during dynamic contractions, suggesting the loss of an energetic "advantage" in older muscle when completing dynamic contractions.

The exact mechanisms responsible for the loss of the energetic "advantage" in older compared with young muscle in response to dynamic contractions is unclear but could be a result of the energy cost of contraction. Metabolic economy (force or power per unit muscle per ATP) is not different between young and older muscle during isometric contractions, but is markedly lower in older muscle during dynamic (both isokinetic and isotonic) contractions [61]. The lower economy and greater ATP cost of dynamic contractions in older compared with younger muscle [44, 45, 61] could be due to an increased energy demand for ion pumping (sodium-potassium, sarcoplasmic reticulum calcium ATPases) [62], myosin ATPases, or increased muscle-tendon unit compliance [63, 64] in older muscle. Regardless of the reason, an elevated cost of contraction could necessitate a greater reliance on non-oxidative pathways to produce the ATP required to meet to energetic demands, resulting in a greater metabolic perturbation in the working muscle. This scenario could explain why the lesser acidosis and accumulation of Pi in response to isometric contractions in older compared with young muscle is eliminated in response to dynamic contractions.

Results from recent meta-analyses evaluating age-related differences in muscle fatigue have concluded that older muscle fatigues to a greater extent than younger muscle when the decline in muscular power is the primary outcome [5, 7]. In contrast, the lack of a difference in intra-myocellular pH or [Pi] between young and older muscle in response to dynamic contractions reported here would predict no age-related difference in fatigue. However, contraction velocity has an important role in the age-related response to dynamic contractions and likely explains the difference between our results and those from the fatigue literature. Callahan and Kent-Braun [8] showed no age-related difference in fatigue in response to an intermediate contraction velocity (individualized to the velocity that elicited ~75% maximal force), but greater fatigue in older compared with young women in response to high-velocity ($270°·s^{-1}$) contractions. This velocity effect may be explained in part by a downward shift in the force-velocity curve of older adults that is exacerbated at high velocities, resulting in greater power decrements in older muscle as the velocity of contraction increases [65].

Variability in the contraction velocities produced in the studies included in the present analysis could explain the lack of difference in pH and [Pi] observed here in response to dynamic contractions, as reflected by the high heterogeneity for these variables (74–81%, Figs 4 and 5). All but one of the studies included here that used a dynamic protocol reported using an isotonic (i.e., constant load) contraction, where velocity is not controlled and often not reported. The one study that did control velocity used a four-min isokinetic contraction protocol at $120°·s^{-1}$ with a significant MD of -0.12 and 2.46 for pH and Pi, respectively, indicating greater acidosis and [Pi] in young compared with older muscle [43]. It is unclear at present how higher velocities would impact these energetic differences, but the current fatigue literature suggests there would be greater acidosis and Pi accumulation in older compared with young muscle.

It is also possible that there are age-related differences in the sensitivity of muscle to the fatigue inducing effects of acidosis and Pi that are dependent on the load and therefore velocity of dynamic contractions. Fitzgerald et al. [43] reported that the slope of the relationship between acidosis and fatigue during isokinetic contractions was less steep in older compared with younger muscle, but not different during isotonic contractions. These results suggest older compared with young muscle is less sensitive to the fatiguing effects of acidosis during high-load, low-velocity contractions, but not during low-load, unconstrained-velocity contractions. In contrast, Sundberg et al. [27] reported no differences in the sensitivity of muscle fibers from young and older men to acidosis and Pi. It is possible that exposing the fibers to saturating levels of calcium, which is not characteristic of fatigue conditions [66, 67], masked possible age-related differences.

## Other potential moderators

Previous meta-analyses quantifying age-related differences in muscle fatigue [5], oxidative capacity [18], and activation [68], have found that the muscle group studied, sex, and habitual physical activity patterns of the participants can significantly influence age-related differences in skeletal muscle energetics and performance. Therefore, it is reasonable to expect that these factors may moderate the age-related difference in muscle acidosis and accumulation of Pi reported here. For example, Larsen et al. [69] found an age-by-muscle interaction for oxidative capacity such that it was greater in the tibialis anterior muscle of older compared with young adults, but lower in the vastus lateralis muscle of the older group; a result that was confirmed in a subsequent systematic review and meta-analysis [18]. Based on the role of oxidative energy production in sparing the accumulation of fatiguing metabolites, it follows that there could be a lower accumulation of metabolites in older compared with young tibialis anterior and the

opposite in the knee extensor muscles. Further, selective atrophy of type II muscle fibers with older age [58] could exacerbate energetic differences between fiber types [60], especially in muscle groups with a large proportion of type II fibers. Similarly, both young [70] and older [10] males have been shown to have a greater contribution to ATP production from non-oxidative energy pathways than females, potentially leading to greater metabolite accumulation and an effect of sex on fatigue.

Lastly, it is well established that health status [71], habitual physical activity [69] and exercise training [72] can influence muscle bioenergetics and fatigue. It is possible that the typical decline in physical activity with older age [73] could influence metabolite accumulation. Additional studies are needed to determine the influence of these factors on age-related differences in intramyocellular acidosis and Pi accumulation.

## Conclusion

This systematic review and meta-analysis provides the first quantitative evaluation of the literature related to potential age-related differences in pH and [Pi] *in vivo* in response to muscular contractions. These data indicate that, overall, older muscle is *more* resistant to the accumulation of these fatigue-inducing metabolites than young muscle, which likely results from the bioenergetic "advantage" of a greater reliance on oxidative energy production in the old. As with the muscle fatigue literature, contraction mode moderates this relationship in that there is greater pH and lower [Pi] in older muscle compared to young muscle in response to isometric contractions but not dynamic contractions. Additional studies are needed to determine whether muscle group, habitual physical activity, or sex moderate the age-related bioenergetic response to contractions, and how these effects may ultimately influence muscle fatigue and mobility in this population.

## Supporting information

**S1 Appendix. Preferred Reporting Items for Systematic Reviews and Meta-Analyses (PRISMA) checklist.** Page numbers reflect the item's location at original submission.
(PDF)

**S2 Appendix. Search strategy.**
(PDF)

**S3 Appendix. Quality assessment scale.**
(PDF)

**S1 Fig. Funnel plot for pH.** The mean difference is plotted on the x-axis and the standard error on the y-axis. Individual studies are represented by gray circles (k = 12, 15 effects). The vertical dashed line represents the overall effect for age-related differences in pH in response to standardized contractions and the diagonal dashed lines represent the 95% confidence interval. Symmetrical distribution of studies and a non-significant Egger's test ($p = 0.446$) suggest there is no publication bias.
(PDF)

**S2 Fig. Funnel plot for inorganic phosphate.** The mean difference is plotted on the x-axis and the standard error on the y-axis. Individual studies are represented by gray circles (k = 9, 12 effects). The vertical dashed line represents the overall effect for age-related differences in [Pi] in response to standardized contractions and the diagonal dashed lines represent the 95% confidence interval. Symmetrical distribution of studies and a non-significant Egger's test

($p$ = 0.122) suggest there is no publication bias.
(PDF)

**S3 Fig. Overall effects for diprotonated inorganic phosphate.** Forest plot for the overall effect of age on intramyocellular diprotonated inorganic phosphate ($H_2PO_4^-$) in response to standardized muscle contractions (k = 5, 8 effects). A positive value represents greater [$H_2PO_4^-$] in young compared with older muscle. Overall, there was no difference in end-exercise [$H_2PO_4^-$] between young and older muscle. However, a sensitivity analysis using the leave-one-out method revealed that when Sundberg et al. [14] was removed, younger muscle had greater intramyocellular [$H_2PO_4^-$] compared with older muscle (MD = 4.23 mM; 95% CI = 2.11, 6.35; $p < 0.01$; $I^2$ = 81%; k = 4; 7 effects).
(PDF)

**S1 Table. Quality assessment scores and risk of bias analysis.** Quality assessment scores for each study from the Newcastle-Ottawa Quality Assessment Scale Modified for Cross-Sectional Studies [7, 37]. This scale assesses the quality of studies based on a total of nine stars across three sections: Selection (four stars), Comparability (two stars), Outcome (three stars). The Risk of Bias analysis rated each study in each section as "good", "fair", or "poor" [38], which are represented as green, yellow, and red, respectively.
(PDF)

## Author Contributions

**Conceptualization:** Luke R. Arieta, Jane A. Kent.

**Formal analysis:** Luke R. Arieta, Zoe H. Smith, Amanda E. Paluch.

**Investigation:** Luke R. Arieta, Zoe H. Smith.

**Methodology:** Luke R. Arieta, Amanda E. Paluch.

**Project administration:** Luke R. Arieta, Jane A. Kent.

**Resources:** Jane A. Kent.

**Software:** Luke R. Arieta, Amanda E. Paluch.

**Supervision:** Jane A. Kent.

**Visualization:** Luke R. Arieta, Jane A. Kent.

**Writing – original draft:** Luke R. Arieta.

**Writing – review & editing:** Zoe H. Smith, Amanda E. Paluch, Jane A. Kent.

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
