## [Decision Letter · Decision Letter 0]

17 Apr 2024

PONE-D-24-07597Effects of old age on contraction-induced intramyocellular acidosis and inorganic phosphate accumulation in vivo: A systematic review and meta-analysisPLOS ONE

Dear Dr. Kent,

Thank you for submitting your manuscript to PLOS ONE. After careful consideration, we feel that it has merit but does not fully meet PLOS ONE’s publication criteria as it currently stands. Therefore, we invite you to submit a revised version of the manuscript that addresses the points raised during the review process.

We look forward to receiving your revised manuscript.

Kind regards,

Daniel Boullosa

Academic Editor

PLOS ONE

Journal Requirements:

Reviewers' comments:

Reviewer's Responses to Questions

**Comments to the Author**

1. Is the manuscript technically sound, and do the data support the conclusions?

Reviewer #1: Yes

Reviewer #2: Yes

2. Has the statistical analysis been performed appropriately and rigorously? 

Reviewer #1: Yes

Reviewer #2: Yes

3. Have the authors made all data underlying the findings in their manuscript fully available?

Reviewer #1: Yes

Reviewer #2: Yes

4. Is the manuscript presented in an intelligible fashion and written in standard English?

Reviewer #1: Yes

Reviewer #2: Yes

5. Review Comments to the Author

Reviewer #1: Attached

Reviewer #2: The present study aimed to systematically review and meta-analyze studies regarding the effects of age (young vs. old) on intramyocellular acidosis and inorganic phosphate accumulation in vivo. The authors focused on cross-sectional responses to standardized skeletal muscle protocols such as dynamic and isometric contraction trials comparing these responses between young (18-45 yr.) and old (55+ yr.) individuals. Overall, the manuscript is well written, the methods and analysis for the systematic review and meta-analysis are adequate to answer the research question and, the conclusions are in line with the obtained results. I have, nevertheless, some minor comments that I would like to address before fully supporting the publication of the manuscript.

1. Abstract:

a. In the abstract, the authors mention “Young muscle acidified more than older muscle”. Although I understand the whole idea, the reality is that is not the muscle that generates the acidification, but the metabolic acidosis is the consequence of the muscle contraction.

b. I am unsure about the last sentence as is not clear. Maybe the reader would benefit from a better characterization of what you define as “an advantage during dynamic contractions”?

2. Introduction:

a. “The way in which fatigue is measured in response to dynamic contractions, either as muscle torque or power, likely contributes to the equivocal results in the literature to date”. Very good point!

b. Regarding the line: “fatigue is measured as the decline in muscle power”, I think fatigue can't be measured. Instead, the decline of muscle power is a proxy measure for a certain type of fatigue.

c. “This greater mitochondrial oxidative capacity of older muscle could allow a relative sparing of the need for non-oxidative ATP production during standardized contraction protocols, and thus reduce acidosis and the accumulation of Pi”. I agree with the premise, however, this would probably depend on the type and intensity of contraction (i.e., maximal vs. submaximal).

3. Methods

a. “Contraction protocols designed to limit acidosis (e.g., short, “oxidative capacity” protocols)”. Please could you elaborate on what you consider oxidative capacity protocols? For instance, what do you consider “short”? In time, would the metabolic pathway dominance depend also on the intensity of muscle contraction?

b. “studies in which mean intramyocellular pH of at least one group was not ≤ 6.90” I am not fully sure about this criterion. Although I understand the rationale behind it and could agree to some degree, I also feel it could be a double-edged sword. In this scenario, the authors state: “Including studies that did not observe muscle pH ≤ 6.90 could artificially mask age-related differences due to an insufficient perturbation from resting” and thus I wonder if the opposite could also be true…

c. Regarding the quality assessment, in addition to the instrument reported, the authors must present the Risk of Bias analysis per included study.

d. After screening the study characteristics in Table 1 one aspect that caught my attention was the diversity of skeletal muscle groups used in the standardized contraction protocols. It is known that there can be variations in inorganic phosphate levels among different skeletal muscle groups and its levels can vary depending on factors such as muscle fiber type composition. For instance, fast-twitch muscle fibers (Type II fibers) typically have higher levels of Pi compared to slow-twitch fibers (Type I fibers). Consequently, muscles with a higher proportion of fast-twitch fibers may exhibit higher Pi concentrations. Additionally, the distribution of Pi can vary within individual muscles based on factors like motor unit recruitment and muscle architecture. Therefore, based on the previous premises I wonder if there could be differences in the outcome of the meta-analysis when using different skeletal muscle groups. I’ve noticed that you mention some of this in the discussion. However, it would be beneficial for the reader if you elaborate on this issues and how may impact on the results you have obtained with the meta-analysis.

4. Discussion

a. In the subsection Age-related differences in pH and [Pi] you mention that “Notably, removing the study by Sundberg et al. [14] resulted in significant overall effects for both pH and [Pi]” However in the results section when referencing the same study you state that “Removing the same study from the overall effect for pH did not change those results”. Please clarify.

b. “This result is in agreement with the overall greater oxidative capacity in older muscle and refutes the concept of a de facto decrease in mitochondrial energy production as a consequence of older age”. Considering the findings of the systematic review and meta-analysis, I believe that employing the term "refutes" may come across as overly assertive. Perhaps opting for a milder verb would better convey the intended meaning.

c. “Older muscle may have lower maximal motor unit discharge rates” I agree, but also older individuals often show reduced fast motor unit and consequently fewer type 2 muscle fibers. The authors discuss this issue in the subsection about “Isometric Contractions” yet, in my opinion, is an issue that deserves being discussed here.

d. “The relatively high (I2 = 82-86%, all studies) heterogeneity in the overall effects reported here is likely due to the inclusion of studies that used different contraction protocols and modes, muscle groups, and participant physical activity status”. This is a pivotal point that warrants a more thorough discussion, given that high values of heterogeny could greatly affect the results of the meta-analysis.

e. In the paragraph about the influence of contraction mode, could you elaborate on the meaning of “energetic advantage”?

6. PLOS authors have the option to publish the peer review history of their article (what does this mean?). If published, this will include your full peer review and any attached files.

Reviewer #1: No

Reviewer #2: **Yes: **Sebastian Del Rosso

---

## [Author Response · Author response to Decision Letter 0]

12 Jun 2024

PONE-D-24-07597

Effects of old age on contraction-induced intramyocellular acidosis and inorganic phosphate accumulation in vivo: A systematic review and meta-analysis

PLOS ONE

Author responses are in italics.

Journal Requirements

We have formatted the manuscript according to the guidelines.

We have added the data for the analysis referred to by this phrase on page 14 and removed the phrase “data not shown.” We confirm that all data reported in this study are now provided in the figures and/or supplemental material.

No retracted studies have been cited. Wilson et al. 1988 has been added to the reference list, as explained in response to Reviewer #1, below. Christie et al. 2014 and 2016, Jubrias et al. 2003, and McPheeters et al. 2012 have been added to the reference list, as explained below in response to Reviewer #2. Fitts 2008 was removed.

Reviewer 1

GENERAL COMMENTS

This systematic review and meta-analysis describes the changes of intracellular pH and inorganic phosphate that occur with fatiguing exercise in aging human muscle versus younger adult muscle. There is also a particular interest in comparing these metabolic changes in static versus dynamic contractions to investigate whether they relate to age-related differences in fatigue resistance with these contraction modes. These are pertinent issues presented with sound rationale. This is a well performed and straightforward piece of work that reads well. My suggestions below should strengthen interest in this work.

We thank Reviewer 1 for their helpful comments and thorough review of the manuscript. We have responded to each comment below.

MAJOR COMMENTS

The authors present mean difference for comparing pHi and Pi between groups. It would be helpful for many readers to also see the absolute values (mean with SD) either in the text or figures. Furthermore, this is important because the same mean pH difference, given as absolute [H+], can be quantitatively different at different pH levels.

The Reviewer is correct in that the same mean difference in pH units can yield different [H+] at different pH levels. For this reason, we have included in Figures 2 and 4 the pH values at the end of the contraction protocols in young and older for each study. This information for Pi appears in Figures 3 and 5. From a statistical standpoint, it would not be appropriate to average the reported means of multiple studies. However, in response to the Reviewer’s comment, we have revised the manuscript to include language in the Results section (page 14) highlighting that mean and SD pH and Pi data for each study are available in the figures.

It would also be extremely valuable for the authors to calculate and evaluate differences in diprotonated Pi between groups given the widespread interest in this species. This would bolster interest in this work.

Diprotonated Pi (H2PO4-) becomes the dominant Pi species below its pKa of 6.75 (see equation, below, Wilson et al. 1988). As such, it reflects changes in both Pi and [H+] and can be considered a “composite” of these two metabolites that can be used to report the metabolic state of the muscle. To our knowledge, H2PO4- has no independent effect on fatigue beyond those mechanisms shown for [Pi] and [H+] (Debold et al 2016, Fitts 2008).

〖[H〗_2 〖PO〗_4^-] =[Pi]/(1+〖10〗^(pH-6.75) )

Of the 12 studies evaluated here, only 5 (8 effects) reported diprotonated Pi, or provided individual participant data to calculate it from pH and Pi. In response to the reviewer’s comment, we have revised the manuscript to include an analysis of this subset of data as a supplement (S3 Fig), and added comments about H2PO4- in the Introduction (page 5) and Discussion (page 17) sections. Again, it would not be statistically appropriate to calculate a mean ± SD for H2PO4- from the means and SDs of pH and Pi reported in the studies included in our meta-analysis, and so we elected not to do that.

Were there any differences in resting pHi and Pi between groups? If so this could indicate different H+ regulating processes at rest.

None of the studies in our analyses reported differences in resting pH or Pi. This information has been added to the text of the Results section (page 14).

Page 15: The Sundberg et al. 2019 study is well-performed (mainly very old female) as is the study by Fitzgerald et al. 2023. The discussion and treatment of these studies, given different responses, needs greater consideration. The difference is postulated to be due to posture differences. Please explain the putative mechanism underpinning this postural effect as it is unclear?

While the reasons for the different results in these two studies is not entirely clear at this point, we believe that it might be related to postural differences. The primary potential mechanism for a effect of postural differences between Sundberg et al. 2019 and Fitzgerald et al. 2023 is that a prone position (as used by Sundberg et al) may compress the feed vessels to the thigh muscles and limit oxygen delivery to the working muscles. A limitation in oxygen availability would necessitate a greater reliance on non-oxidative ATP production with subsequent Pi accumulation and acidosis. For example, in Lanza et al. 2007, older muscle had lower glycolytic flux during normal blood flow isometric contraction compared with young muscle, but when blood flow was occluded the older muscle increased their glycolytic ATP production to equal that of young muscle. During dynamic contractions, which are more energetically costly compared with isometric contractions, these results may be exacerbated and potentially explain the different results between Sundberg et al. 2019 and Fitzgerald et al. 2023. In response to the reviewer’s comment, we have revised the discussion to include this point (page 18). 

MINOR COMMENTS

The authors should briefly acknowledge other metabolic changes (and other fatigue mechanisms) that are likely to occur during the exercise protocols of the present review.

As suggested by the reviewer, we have revised our Introduction (page 5) to include mention of other metabolic changes (ATP, glycogen) and fatigue mechanisms, including excitation-contraction coupling failure and central fatigue, that can occur during contraction protocols such as reported here (Kent-Braun et al. 2012).

The authors interchange use of intramyocellular and intracellular – it would be better to consistently use one expression.

We thank the reviewer for pointing out this potential source of confusion and have changed all occurrences to “intramyocellular.”

Reviewer 2

The present study aimed to systematically review and meta-analyze studies regarding the effects of age (young vs. old) on intramyocellular acidosis and inorganic phosphate accumulation in vivo. The authors focused on cross-sectional responses to standardized skeletal muscle protocols such as dynamic and isometric contraction trials comparing these responses between young (18-45 yr.) and old (55+ yr.) individuals. Overall, the manuscript is well written, the methods and analysis for the systematic review and meta-analysis are adequate to answer the research question and, the conclusions are in line with the obtained results. I have, nevertheless, some minor comments that I would like to address before fully supporting the publication of the manuscript.

We thank reviewer 2 for their careful review and valuable feedback on our manuscript. We have responded to each comment below.

Abstract:

In the abstract, the authors mention “Young muscle acidified more than older muscle”. Although I understand the whole idea, the reality is that is not the muscle that generates the acidification, but the metabolic acidosis is the consequence of the muscle contraction.

We agree that the intramyocellular acidosis studied here is a consequence of the increased energy demand due to muscular contraction, and have revised the abstract to clarify this point.

I am unsure about the last sentence as is not clear. Maybe the reader would benefit from a better characterization of what you define as “an advantage during dynamic contractions”?

We have revised the text to clarify that the “advantage” we are referring to is a greater use of oxidative phosphorylation and, consequently, a lower accumulation of fatiguing metabolites during contractions.

Introduction:

“The way in which fatigue is measured in response to dynamic contractions, either as muscle torque or power, likely contributes to the equivocal results in the literature to date”. Very good point!

Thank you.

Regarding the line: “fatigue is measured as the decline in muscle power”, I think fatigue can't be measured. Instead, the decline of muscle power is a proxy measure for a certain type of fatigue.

This sentence refers to papers that have operationally defined fatigue as a decline in muscle power. In response to this comment, we have revised this sentence to reflect this point: “…when muscle fatigue is operationally defined and measured as the decline in power…” in order to be more precise (page 4).

“This greater mitochondrial oxidative capacity of older muscle could allow a relative sparing of the need for non-oxidative ATP production during standardized contraction protocols, and thus reduce acidosis and the accumulation of Pi”. I agree with the premise, however, this would probably depend on the type and intensity of contraction (i.e., maximal vs. submaximal).

We agree and have added, “particularly during high-intensity contractions that likely recruit all fiber types” to the end of this sentence to clarify and added Christie et al. 2014 and 2016 to support this statement (page 5).

Methods

“Contraction protocols designed to limit acidosis (e.g., short, “oxidative capacity” protocols)”. Please could you elaborate on what you consider oxidative capacity protocols? For instance, what do you consider “short”? In time, would the metabolic pathway dominance depend also on the intensity of muscle contraction?

Thank you for this comment and the opportunity to clarify. “Oxidative capacity protocols” are traditionally short (e.g., 24 s) in order to decrease [PCr] by approximately 50% and limit intramyocellular acidosis, in accordance with best practices for measuring muscle oxidative capacity (Meyer 1988, Jubrias et al. 2003, Meyerspeer et al. 2021). We did not eliminate any studies based purely on the length of the contraction protocol; rather, studies were eliminated if the protocol was specifically designed to limit acidosis, and then successfully did so. This point, which works in tandem with the cutoff for pH of 6.9 (addressed below) has been clarified in the Methods (page 8). We agree that metabolic pathway dominance could also depend in part on the intensity of muscle contraction, and have revised the Discussion to include mention of contraction intensity and concomitant motor unit recruitment behavior (page 19). 

“studies in which mean intramyocellular pH of at least one group was not ≤ 6.90” I am not fully sure about this criterion. Although I understand the rationale behind it and could agree to some degree, I also feel it could be a double-edged sword. In this scenario, the authors state: “Including studies that did not observe muscle pH ≤ 6.90 could artificially mask age-related differences due to an insufficient perturbation from resting” and thus I wonder if the opposite could also be true…

A cutoff for muscle acidosis has previously been used at pH < 6.9 (Jubrias et al. 2003), and is consistent with the recommendations in the recent consensus paper for using 31P MRS in skeletal muscle (Meyerspeer et al. 2021). We have clarified in the methods that this cutoff level of 6.9 pH was selected, in part, based on this previously-established level. The significance of this 6.9 level is that oxidative capacity may be slowed when pH drops below it. Therefore, numerous studies with the purpose of measuring oxidative capacity have implemented protocols to limit the amount of muscle acidosis. Potential age-related differences in pH are therefore purposefully dampened and inclusion of those studies in this meta-analysis would saturate the overall effect towards no difference by age. We have revised the manuscript to include these points and references (page 8). 

Regarding the quality assessment, in addition to the instrument reported, the authors must present the Risk of Bias analysis per included study.

In response to this point, we have scored the risk of bias for each study as “good”, “fair”, or “poor” based on an established scoring chart (McPheeters et al. 2012; https://www.ncbi.nlm.nih.gov/books/NBK107322/), and updated the Methods (page 10), Results (page 16), and S1 Table, to reflect this additional information.

After screening the study characteristics in Table 1 one aspect that caught my attention was the diversity of skeletal muscle groups used in the standardized contraction protocols. It is known that there can be variations in inorganic phosphate levels among different skeletal muscle groups and its levels can vary depending on factors such as muscle fiber type composition. For instance, fast-twitch muscle fibers (Type II fibers) typically have higher levels of Pi compared to slow-twitch fibers (Type I fibers). Consequently, muscles with a higher proportion of fast-twitch fibers may exhibit higher Pi concentrations. Additionally, the distribution of Pi can vary within individual muscles based on factors like motor unit recruitment and muscle architecture. Therefore, based on the previous premises I wonder if there could be differences in the outcome of the meta-analysis when using different skeletal muscle groups. I’ve noticed that you mention some of this in the discussion. However, it would be beneficial for the reader if you elaborate on this issues and how may impact on the results you have obtained with the meta-analysis.

We agree that the skeletal muscle studied may impact the results and we have elaborated on our discussion of this point with regard to fiber type composition in the “Other potential moderators” section (page 24).

Discussion

In the subsection Age-related differences in pH and [Pi] you mention that “Notably, removing the study by Sundberg et al. [14] resulted in significant overall effects for both pH and [Pi]” However in the results section when referencing the same study you state that “Removing the same study from the overall effect for pH did not change those results”. Please clarify.

We apologize for the confusion and have revised

---

## [Decision Letter · Decision Letter 1]

5 Jul 2024

PONE-D-24-07597R1Effects of old age on contraction-induced intramyocellular acidosis and inorganic phosphate accumulation in vivo: A systematic review and meta-analysisPLOS ONE

Dear Dr. Kent,

Thank you for submitting your manuscript to PLOS ONE. After careful consideration, we feel that it has merit but does not fully meet PLOS ONE’s publication criteria as it currently stands. Therefore, we invite you to submit a revised version of the manuscript that addresses the points raised during the review process. Please, consider the suggestions by Reviewer #2 before acceptance.

We look forward to receiving your revised manuscript.

Kind regards,

Daniel Boullosa

Academic Editor

PLOS ONE

Journal Requirements:

Reviewers' comments:

Reviewer's Responses to Questions

**Comments to the Author**

1. If the authors have adequately addressed your comments raised in a previous round of review and you feel that this manuscript is now acceptable for publication, you may indicate that here to bypass the “Comments to the Author” section, enter your conflict of interest statement in the “Confidential to Editor” section, and submit your "Accept" recommendation.

Reviewer #1: (No Response)

Reviewer #2: All comments have been addressed

2. Is the manuscript technically sound, and do the data support the conclusions?

Reviewer #1: Yes

Reviewer #2: Yes

3. Has the statistical analysis been performed appropriately and rigorously? 

Reviewer #1: Yes

Reviewer #2: Yes

4. Have the authors made all data underlying the findings in their manuscript fully available?

Reviewer #1: Yes

Reviewer #2: Yes

5. Is the manuscript presented in an intelligible fashion and written in standard English?

Reviewer #1: Yes

Reviewer #2: Yes

6. Review Comments to the Author

Reviewer #1: GENERAL COMMENTS

As mentioned previously this systematic review and meta-analysis describes the changes of intracellular pH and inorganic phosphate that occur with fatiguing exercise in aging human muscle versus younger adult muscle, which are pertinent issues presented with sound rationale. The further consideration of H2PO4- was a helpful addition – thanks for including this. Overall, this is a well-performed piece of work and the tweaking of the text has been beneficial

COMMENTS

As mentioned previously it would be “helpful” for many readers to also see the absolute values for pHi and Pi between groups (mean with SD) either in the text or figures. This has been argued against. I agree that readers can see the data from the individual studies, and also calculate the mean data across studies from this. The authors need to decide whether they want to be “helpful” or not? Comments “from a statistical standpoint…” do not hold if the authors also show the mean difference for these values which then has statistical issues.

The effects of pH and inorganic phosphate on force production appear to be synergistic rather than additive. Like you mention much of the Pi exists as H2PO4- under acidic milieu. Hence the reason to evaluate H2PO4- in young and older muscle, regardless of whether or not it is simply total Pi or H2PO4- that contributes to fatigue mechanisms.

Introduction – the authors have now mentioned other potential mechanisms of fatigue (page 5) although it would also be helpful to mention electrolyte shifts and reactive oxygen species (perhaps refer to a review). The comment that these other aspects will contribute to muscle fatigue by a smaller extent than pH and Pi is questionable – there is no doubt that pH and Pi are important bioenergetic contributors to fatigue but quantitative comments have not been justified.

It may be wiser to use “older age” rather than “old age” in the title, even given that the authors found no statistical difference for those <65 yr versus the more elderly. For authors consideration only.

Reviewer #2: I would like to express my gratitude and acknowledge the authors' efforts in addressing all of my comments. All the issues I raised have been thoroughly resolved, and as a result, I recommend the manuscript for publication.

7. PLOS authors have the option to publish the peer review history of their article (what does this mean?). If published, this will include your full peer review and any attached files.

Reviewer #1: No

Reviewer #2: **Yes: **Sebastián Del Rosso

---

## [Author Response · Author response to Decision Letter 1]

17 Jul 2024

PONE-D-24-07597

Effects of old age on contraction-induced intramyocellular acidosis and inorganic phosphate accumulation in vivo: A systematic review and meta-analysis

PLOS ONE

Reviewer #1: GENERAL COMMENTS

As mentioned previously this systematic review and meta-analysis describes the changes of intracellular pH and inorganic phosphate that occur with fatiguing exercise in aging human muscle versus younger adult muscle, which are pertinent issues presented with sound rationale. The further consideration of H2PO4- was a helpful addition – thanks for including this. Overall, this is a well-performed piece of work and the tweaking of the text has been beneficial

COMMENTS

As mentioned previously it would be “helpful” for many readers to also see the absolute values for pHi and Pi between groups (mean with SD) either in the text or figures. This has been argued against. I agree that readers can see the data from the individual studies, and also calculate the mean data across studies from this. The authors need to decide whether they want to be “helpful” or not? Comments “from a statistical standpoint…” do not hold if the authors also show the mean difference for these values which then has statistical issues.

We have added the mean and SD for the absolute values for pH, Pi, and H2PO4- to the Results section. These have been calculated according to the equation in Table 6.5.a in the Cochrane Handbook (https://training.cochrane.org/handbook/current/chapter-06#section-6-5). This information has been added to the methods section. 

The effects of pH and inorganic phosphate on force production appear to be synergistic rather than additive. Like you mention much of the Pi exists as H2PO4- under acidic milieu. Hence the reason to evaluate H2PO4- in young and older muscle, regardless of whether or not it is simply total Pi or H2PO4- that contributes to fatigue mechanisms

Introduction – the authors have now mentioned other potential mechanisms of fatigue (page 5) although it would also be helpful to mention electrolyte shifts and reactive oxygen species (perhaps refer to a review). The comment that these other aspects will contribute to muscle fatigue by a smaller extent than pH and Pi is questionable – there is no doubt that pH and Pi are important bioenergetic contributors to fatigue but quantitative comments have not been justified.

We have added mention of electrolytes and reactive oxygen species to the other potential mechanisms, which are covered in reference 16. We have also removed the comment “although likely to a smaller extent than pH and Pi” to avoid any quantitative comments.

It may be wiser to use “older age” rather than “old age” in the title, even given that the authors found no statistical difference for those <65 yr versus the more elderly. For authors consideration only. 

We have made this change, as requested.

Reviewer #2: I would like to express my gratitude and acknowledge the authors' efforts in addressing all of my comments. All the issues I raised have been thoroughly resolved, and as a result, I recommend the manuscript for publication.

Thank you.

---

## [Decision Letter · Decision Letter 2]

23 Jul 2024

Effects of older age on contraction-induced intramyocellular acidosis and inorganic phosphate accumulation in vivo: A systematic review and meta-analysis

PONE-D-24-07597R2

Dear Dr. Kent,

We’re pleased to inform you that your manuscript has been judged scientifically suitable for publication and will be formally accepted for publication once it meets all outstanding technical requirements.

Kind regards,

Daniel Boullosa

Academic Editor

PLOS ONE

Additional Editor Comments (optional):

Reviewers' comments:

Reviewer's Responses to Questions

**Comments to the Author**

1. If the authors have adequately addressed your comments raised in a previous round of review and you feel that this manuscript is now acceptable for publication, you may indicate that here to bypass the “Comments to the Author” section, enter your conflict of interest statement in the “Confidential to Editor” section, and submit your "Accept" recommendation.

Reviewer #1: All comments have been addressed

Reviewer #2: All comments have been addressed

2. Is the manuscript technically sound, and do the data support the conclusions?

Reviewer #1: Yes

Reviewer #2: Yes

3. Has the statistical analysis been performed appropriately and rigorously? 

Reviewer #1: Yes

Reviewer #2: Yes

4. Have the authors made all data underlying the findings in their manuscript fully available?

Reviewer #1: Yes

Reviewer #2: Yes

5. Is the manuscript presented in an intelligible fashion and written in standard English?

Reviewer #1: Yes

Reviewer #2: (No Response)

6. Review Comments to the Author

Reviewer #1: Congratulations to the authors who have done a great job in addressing all the amendments needed to enhance the quality of this review article.

Reviewer #2: Again, I would like to express my gratitude and acknowledge the authors' efforts in addressing all of my comments. All the issues I raised have been thoroughly resolved, and as a result, I recommend the manuscript for publication.

7. PLOS authors have the option to publish the peer review history of their article (what does this mean?). If published, this will include your full peer review and any attached files.

Reviewer #1: No

Reviewer #2: No

---

## [Editor Report · Acceptance letter]

7 Aug 2024

PONE-D-24-07597R2

PLOS ONE

Dear Dr. Kent,

I'm pleased to inform you that your manuscript has been deemed suitable for publication in PLOS ONE. Congratulations! Your manuscript is now being handed over to our production team.

Kind regards,

on behalf of

Dr. Daniel Boullosa

Academic Editor

PLOS ONE